# Metabolomics Reveals Tyrosine Kinase Inhibitor Resistance-Associated Metabolic Events in Human Metastatic Renal Cancer Cells

**DOI:** 10.3390/ijms25126328

**Published:** 2024-06-07

**Authors:** Filipa Amaro, Márcia Carvalho, Maria de Lourdes Bastos, Paula Guedes de Pinho, Joana Pinto

**Affiliations:** 1Associate Laboratory i4HB-Institute for Health and Bioeconomy, University of Porto, 4050-313 Porto, Portugal; mcarv@ufp.edu.pt (M.C.); mlbastos@ff.up.pt (M.d.L.B.); pguedes@ff.up.pt (P.G.d.P.); 2UCIBIO-Applied Molecular Biosciences Unit, Laboratory of Toxicology, Department of Biological Sciences, Faculty of Pharmacy, University of Porto, 4050-313 Porto, Portugal; 3RISE-UFP, Health Research Network, Faculty of Health Sciences, University Fernando Pessoa, 4200-150 Porto, Portugal

**Keywords:** renal cell carcinoma, tyrosine kinase inhibitors, metabolic reprogramming, drug resistance, metabolomics

## Abstract

The development of resistance to tyrosine kinase inhibitors (TKIs) is a major cause of treatment failure in metastatic renal cell carcinoma (mRCC). A deeper understanding of the metabolic mechanisms associated with TKI resistance is critical for refining therapeutic strategies. In this study, we established resistance to sunitinib and pazopanib by exposing a parental Caki-1 cell line to increasing concentrations of sunitinib and pazopanib. The intracellular and extracellular metabolome of sunitinib- and pazopanib-resistant mRCC cells were investigated using a nuclear magnetic resonance (NMR)-based metabolomics approach. Data analysis included multivariate and univariate methods, as well as pathway and network analyses. Distinct metabolic signatures in sunitinib- and pazopanib-resistant RCC cells were found for the first time in this study. A common metabolic reprogramming pattern was observed in amino acid, glycerophospholipid, and nicotinate and nicotinamide metabolism. Sunitinib-resistant cells exhibited marked alterations in metabolites involved in antioxidant defence mechanisms, while pazopanib-resistant cells showed alterations in metabolites associated with energy pathways. Sunitinib-resistant RCC cells demonstrated an increased ability to proliferate, whereas pazopanib-resistant cells appeared to restructure their energy metabolism and undergo alterations in pathways associated with cell death. These findings provide potential targets for novel therapeutic strategies to overcome TKI resistance in mRCC through metabolic regulation.

## 1. Introduction

The incidence of renal cell carcinoma (RCC) has been steadily increasing, making it the third most common urological malignancy [1,2]. Clear cell RCC (ccRCC) is the most common histological subtype among all RCC subtypes, accounting for over 75% of diagnosed cases [3]. In the early stages, ccRCC is effectively treated with surgery (partial or radical nephrectomy); however, in advanced or metastatic cases, systemic therapies are required to reduce the risk of recurrence [4]. Conventional chemotherapy and radiotherapy have shown limited efficacy in advanced disease, prompting research efforts into new and innovative therapeutic options, such as targeted therapy and immunotherapy [5].

The Von Hippel Lindau (*VHL*) gene inactivation is a well-established hallmark of ccRCC, leading to increased vascularisation [6]. This abnormality increases the expression of several hypoxia-inducible factors and pro-angiogenic factors, namely vascular endothelial and platelet-derived growth factors (VEGF and PDGF, respectively), which contribute to ccRCC development and progression [7]. Indeed, the therapeutic landscape for metastatic ccRCC has changed with the introduction of anti-angiogenic agents, in particular tyrosine kinase inhibitors (TKIs) [8]. These agents have important advantages, including low toxicity and high specificity, and can be used in combination with other therapeutic agents [9]. Among the various approved TKIs, sunitinib and pazopanib are widely preferred as first-line treatments for metastatic ccRCC (mRCC) [3,10]. Sunitinib is an orally administered TKI with antitumor and anti-angiogenic activity. These effects are mediated by the inhibition of VEGF receptors (VEGFRs) as well as the platelet-derived growth factor receptors (PDGFRs) α and β, and c-Kit kinase [11]. With a similar mechanism of action, pazopanib effectively targets VEGFRs, PDGFRs, c-Kit, as well as fibroblast growth factor receptors (FGFRs), IL-2-inducible T-cell kinase, and lymphocyte-specific protein tyrosine kinase [12,13].

mRCC patients treated with sunitinib and pazopanib experience improved progression-free and overall survival; however, resistance typically develops within 6–12 months [3,14]. The underlying mechanisms of TKI resistance are not fully understood, but the available studies consistently point to the unique and heterogeneous environment within ccRCC, characterised by hypoxia, as an obstacle to the efficacy of these drugs [15,16]. Hypoxia induces epithelial-mesenchymal transition (EMT), which increases the ability of cells to sequester TKIs and also triggers alternative angiogenic pathways [9,17]. These include the increased expression of proangiogenic factors and the initiation of non-angiogenic pathways, such as vascular co-option—a mechanism by which cancer cells use the existing vasculature for growth support [15,16]. Lysosomal sequestration, characterised by the accumulation of TKIs within the lysosomal compartment preventing the drug from reaching its target [5], is strongly associated with sunitinib resistance [18,19,20]. In addition, non-coding RNA mutations and the presence of single nucleotide polymorphisms are recognised contributors to TKI resistance as they modulate tumour suppressor genes [11]. A deeper understanding of the alterations associated with TKI resistance is of considerable importance in restoring cancer cell responsiveness to therapy and thus improving the prognosis of metastatic ccRCC patients.

Metabolomics provides a convenient strategy to investigate the changes in metabolite levels that occur in resistant cancer cells and contribute to tumour progression [21,22]. Only two studies have reported metabolic alterations associated with sunitinib resistance [18,23]. The main alterations were in the energy metabolism and antioxidant capacity of sunitinib-resistant cells. However, there is currently a lack of studies investigating the metabolic changes associated with pazopanib resistance. In our recent work, we investigated the metabolic response of ccRCC cells to sunitinib and pazopanib [24]. By combining traditional toxicity assessments and metabolomics, we found important differences in the cellular and metabolic effects induced by each TKI on ccRCC cells. Key findings suggest that sunitinib has superior tumour selectivity and lower nephrotoxic potential compared to pazopanib. Therefore, it is essential to thoroughly investigate the resistance mechanisms associated with each TKI. The present study uses a metabolomics approach to assess, for the first time to our knowledge, both intracellular and extracellular metabolic perturbations associated with sunitinib and pazopanib resistance in metastatic ccRCC cells. To this end, we established sunitinib- and pazopanib-resistant RCC cell sublines from the same parental cell line. We then performed a comprehensive analysis of intracellular (polar and lipid) and extracellular (present in the culture medium) metabolites of parental and TKI-resistant cells using proton (^1^H) nuclear magnetic resonance (NMR) spectroscopy-based metabolomics. The study identified several metabolic changes associated with resistance to sunitinib and pazopanib. These findings may contribute to a better understanding of the metabolic reprogramming of TKI-resistant metastatic ccRCC cells. Targeting these metabolic dysregulations may be a promising strategy to overcome the development of sunitinib and pazopanib resistance in mRCC patients.

## 2. Results

### 2.1. Establishment of Sunitinib- and Pazopanib-Resistant Renal Cancer Cell Lines

Sunitinib and pazopanib are commonly used as first-line drugs for advanced ccRCC. In our study, we chose the Caki-1 cell line, which is known for its metastatic potential, to mimic in vivo conditions. To confirm the successful establishment of resistant cell lines, we first evaluated the sensitivity of the parental and resistant Caki-1 cell lines to sunitinib and pazopanib. Figure 1a shows the concentration-response curves of the established sunitinib-resistant vs. parental Caki-1 cells when exposed to a wide range of sunitinib concentrations (0.1–100 µM), while Figure 1b shows the concentration-response curves of the pazopanib-resistant vs. parental Caki-1 cells when treated with pazopanib (0.1–200 µM). Note that a full sigmoidal curve was not achieved for pazopanib as we were unable to test pazopanib concentrations above 200 µM due to solubility limitations. The EC_50_ values for sunitinib in parental and sunitinib-resistant Caki-1 cells were 4.7 and 17.6 µM, respectively, whereas the EC_50_ values for pazopanib in parental and pazopanib-resistant Caki-1 cells were 51.6 and 175.6 µM, respectively. Based on these results, sunitinib- and pazopanib-resistant Caki-1 cells were approximately 3.8- and 3.4-fold more resistant to drug treatment than their parental cells, respectively.

The resistance behaviour of the cells was further assessed in the proliferation assay. Figure 1c shows a statistically significant increase in proliferation of sunitinib-resistant Caki-1 cells after 48 and 72 h of exposure to sunitinib (2 µM) compared to parental Caki-1 cells. However, no statistically significant difference in parental vs. pazopanib-resistant Caki-1 cell growth was observed (Figure 1d). Overall, our results demonstrate that Caki-1 cells become resistant to sunitinib and, to a lesser extent, to pazopanib.

Finally, the resistant cell sublines were characterised by unique morphological changes that were more pronounced in sunitinib-resistant Caki-1 cells, such as enlarged cytoplasm and severe vacuolisation (Figure 1e). These cells also exhibited a bright yellow colour that is characteristic of sunitinib, suggesting that this drug accumulates intracellularly in resistant cells.

### 2.2. H NMR Metabolic Profiles of Caki-1 Cells and Culture Medium

Representative ^1^H NMR spectra of the intracellular polar extract and extracellular culture medium of Caki-1 cells are shown in Appendix A, respectively. A total of 24 polar metabolites were identified in the intracellular polar extract of Caki-1 cells, including various amino acids and derivatives, organic acids, phosphocholines, nucleotides and sugars. Metabolic profiling of the extracellular culture medium of Caki-1 cells resulted in the detection of 25 metabolites, including several amino acids, organic acids, and sugars. Appendix A provide a comprehensive annotation of the metabolites that were detected in both profiles, and Appendix A illustrates the number of unique and shared metabolites between them. Regarding the intracellular lipidome, Appendix A shows a representative ^1^H NMR spectrum of the intracellular lipid extract of Caki-1 cells. A total of seven lipid species were annotated, including cholesterol, cholesteryl esters, fatty acids, phosphatidylethanolamines, phosphatidylcholines, monoglycerides and triglycerides, as described in Appendix A. STOCSY allowed the identification of more resolved spin systems for monoglycerides and triglycerides, as shown in Appendix A.

### 2.3. Metabolic Changes in Sunitinib- and Pazopanib-Resistant Caki-1 Cells

To identify the metabolic perturbations associated with sunitinib and pazopanib resistance, the ^1^H NMR data matrices of the intracellular and extracellular metabolic profiles of Caki-1 cells were subjected to multivariate analysis. The principal component analysis (PCA) models obtained revealed no apparent separation between resistant and parental cells in either intracellular or extracellular metabolic profiles. On the other hand, partial least squares-discriminant analysis (PLS-DA) differentiated the intracellular polar and lipid contents of the two resistant cell lines from the parental cell line, as shown in the PLS-DA score scatter plots in Figure 2a,c and Figure 3a,c for sunitinib- and pazopanib-resistant cells, respectively. The predictive ability (Q^2^) of these models (ranging from 0.39 to 0.73), obtained through cross-validation, supported the discrimination between groups. Clear discriminations were also obtained for the extracellular metabolic profiles of resistant and parental cells in PLS-DA models (Figure 2e and Figure 3e) with predictive abilities (Q^2^) ranging from 0.46 to 0.55. The intracellular and extracellular metabolites responsible for model discrimination were interpreted in the loading plots shown in Figure 2b,d,f and Figure 3b,d,f. The metabolites with variable importance in the projection (VIP) greater than 1 in the PLS-DA loading plots that varied significantly according to univariate analysis were plotted in the heatmaps shown in Figure 4a,b. The effect size and *p*-value of these metabolites can be found in Appendix A. Overall, sunitinib-resistant Caki-1 cells showed statistically significant changes in 15 intracellular and two extracellular metabolites. Pazopanib-resistant cells showed a total of 15 intracellular and 17 extracellular significantly altered metabolites. In particular, sunitinib-resistant Caki-1 cells were characterised by a significant intracellular increase in the levels of several amino acids and derivatives (alanine, aspartate, glycine, glutathione, isoleucine, leucine, and valine), ethanolamine, lipid species (cholesteryl esters, monoglycerides and phosphatidylethanolamines) and myo-inositol. An intracellular decrease in phosphocholine, glycerophosphocholine and NAD^+^ was also observed. The phosphocholine/glycerophosphocholine (ChoP/GPC) ratio was also calculated (Appendix A) as an indicator of cancer cell growth in the resistant cell lines compared to the parental cells. Indeed, sunitinib-resistant cells showed a significantly higher ChoP/GPC ratio compared to parental cells, whereas no significant changes were observed in pazopanib-resistant cells. Analysis of the culture medium without cells (blanks) was a useful strategy to evaluate variations in metabolite excretion (E) or consumption (C), as shown in Appendix A and the boxplots shown in Appendix A. Indeed, the extracellular metabolic changes in sunitinib resistance included only a significant decrease in glutamine levels (higher consumption) and a significant increase in lactate levels (higher excretion) compared to parental cells.

The intracellular and extracellular metabolic profiles of pazopanib-resistant cells were mostly different from those observed in sunitinib-resistant cells. Common alterations included a significant intracellular increase in the levels of some amino acids (aspartate, alanine, isoleucine, leucine, glycine), monoglycerides and a significant decrease in phosphocholines (ChoP and GPC) and NAD^+^. Specific changes in the intracellular profile of pazopanib-resistant cells included a significant increase in lactate levels and a significant decrease in taurine, total fatty acids, unsaturated fatty acids, cholesterol, and cholesteryl esters. The extracellular metabolic profile of pazopanib-resistant cells differs from sunitinib-resistant cells in the number of significantly altered metabolites compared to parental cells. Specific alterations observed in pazopanib-resistant Caki-1 cells included a significant decrease in the extracellular levels of several amino acids (alanine, arginine, asparagine, aspartate, glycine, glutamine, isoleucine, lysine, methionine, tyrosine, and valine) and organic acids (acetate, formate, fumarate, pyruvate, and succinate) and a significant increase in extracellular glucose and aspartate. Interestingly, these changes reflected a lower glucose consumption, higher amino acid consumption (except for alanine and aspartate) and lower organic acid excretion (except for succinate), as can be seen by comparison with the blank (Appendix A).

The comprehensive integration of the significantly altered metabolites into pathways suggested major changes in amino acid, glycerophospholipid, and nicotinate and nicotinamide metabolism in both resistant cell lines (Figure 4a,b). Sunitinib-resistant cells were also characterised by putative dysregulations in glutathione and inositol phosphate metabolism, whereas pazopanib-resistant cells showed putative dysregulations in glycolysis/gluconeogenesis, tricarboxylic acid (TCA) cycle, pyruvate metabolism, and lipid metabolism. In addition, Figure 4c and Figure 4d show a summary plot of the pathway analysis performed for sunitinib- and pazopanib-resistant cells, respectively. This analysis ranks metabolite sets based on their associated *p*-values and pathway impact values, providing insight into the most potentially dysregulated metabolic pathways involved in sunitinib and pazopanib resistance. A total of five metabolic pathways were found to be significantly altered in sunitinib-resistant cells, including amino acid, glycerophospholipid and glutathione metabolism. Pathway analysis of pazopanib-resistant cells showed eight statistically significant alterations in several amino acid and energy metabolic pathways.

Network analysis, a bioinformatics tool that allows the exploration of interconnected metabolites and genes, was also used to identify putative genes associated with the metabolic dysregulations induced by TKI resistance. The detailed results of this analysis are presented in Table 1, and the interaction network visualisations are shown in Appendix A for sunitinib- and pazopanib-resistant cell lines, respectively. Regarding sunitinib-resistant Caki-1, notable connections (illustrated by the higher nodes in Appendix A) are observed between different amino acids and genes responsible for amino acid synthesis (e.g., *IARS*, *KARS* and *EPRS*) and transport (e.g., *SLC7A8*, *SLC38A5* and *SLC38A2*). Similarly, the putative perturbations observed in pazopanib-resistant cells (Appendix A) suggested alterations in genes associated with amino acid metabolism and those encoding mechanisms related to energy processes (e.g., *PKMS*, *PKLR* and *LDHAs*).

## 3. Discussion

### 3.1. Successful Establishment of Sunitinib- and Pazopanib-Resistant mRCC Cell Lines

Cancer cells reprogram their metabolism to sustain tumour progression and develop resistance to anticancer therapies [21]. Therefore, changes in the cancer cell metabolome, translated by changes in metabolite levels, may specifically reflect the onset and development of therapeutic resistance. In this work, we established sunitinib- and pazopanib-resistant RCC cell lines derived from a parental mRCC cell line to investigate the metabolic changes underlying the development of resistance to each drug. Previous studies have outlined protocols for establishing sunitinib-resistant cell lines [18,23,25,26,27], but there is a lack of studies focusing on pazopanib resistance in RCC cells. In the present work, continuous exposure to each TKI resulted in a 3.8- and 3.4-fold increase in the EC_50_ values for sunitinib- and pazopanib-resistant cells, respectively, compared to the parental cells (Figure 1). In the literature, this fold change is commonly referred to as the resistance factor [20,28]. For the successful development of resistant cell lines, a resistance factor greater than 2.5 is considered [20,28], which is consistent with our results. However, it should be noted that the acquired resistance was more pronounced in metastatic ccRCC cells that were continuously exposed to sunitinib than to pazopanib. In addition to the higher resistance factor, sunitinib-resistant cells showed an impressively higher proliferation rate than parental cells (Figure 1c), as well as substantial morphological abnormalities such as enlarged cytoplasm with extensive vacuolization (Figure 1e). This heterogeneity in the cell body has also been reported by other authors, who suggested that it may be an adaptation of the cells to redistribute drugs [27]. In addition, sunitinib-resistant cells exhibited the characteristic yellow colour of sunitinib, consistent with previous reports of higher sunitinib levels in the lysosomal compartment of resistant RCC cells compared to parental cells [18,19,20]. ATP-binding cassette (ABC) transporters are responsible for mediating the sequestration process that increases the capacity of cancer cells to store sunitinib [9,11]. Targeting these transporters has emerged as a strategy to reverse lysosomal drug sequestration. This strategy aims to restore the efficacy of sunitinib, as demonstrated by Gotink et al. [20], while also offering a potential approach to overcome drug resistance [29].

### 3.2. Metabolic Events behind Sunitinib and Pazopanib Resistance in RCC Cells

First, it should be noted that previous metabolomic studies investigating metabolic changes associated with sunitinib resistance have only examined the intracellular metabolome [18,23]. The current work integrates the analysis of both intracellular (including polar and lipid extracts) and extracellular metabolomes of resistant cells. Figure 5 provides an overview of the putative intracellular and extracellular metabolic shifts found in sunitinib- and pazopanib-resistant RCC cell lines in this study.

Overall, the common metabolic changes associated with resistance to these TKIs were identified mainly at the intracellular level, including an up-regulation of several amino acids (aspartate, alanine, isoleucine, leucine, and glycine), while only an increased glutamine consumption was observed at the extracellular level. Amino acids are basic units of proteins that have structural functions and fundamental roles in maintaining cellular homeostasis, energy production, and redox balance [21,30]. These findings suggest a common regulatory mechanism in several metabolic pathways related to amino acid metabolism (aminoacyl-tRNA biosynthesis, branched-chain amino acid biosynthesis and degradation and alanine, aspartate, and glutamate metabolism) in resistant cells to ensure their survival and growth. Consistent with these results, a previous study conducted by Hatakeyama et al. [18] investigating metabolic changes in a sunitinib-resistant RCC cell line (786-O cell line) showed a significant increase in the intracellular levels of several amino acids, including glycine, tyrosine, valine, isoleucine, aspartate, and glutamate. The study conducted by Sato et al. [23], investigating metabolic changes associated with sunitinib resistance in tumour tissue from RCC 786-O injected mice, showed an intracellular increase in glutamine levels. This increase was supported by a significant overexpression of glutamine transporters, such as *SLC145*. Results from our network analysis also suggested a putative association with several *SLC* genes, a family of membrane-transporters that mediate the flux of several nutrients and metabolites, including amino acids. Taken together, these results establish a link between increased cellular glutamine uptake as a key mechanism underlying sunitinib resistance in ccRCC. Notably, glutamine addiction (i.e., increased demand for glutamine) and increased expression of glutaminase (an enzyme that converts glutamine to glutamate) represent distinct metabolic alterations in ccRCC and appear to be exacerbated in the context of sunitinib resistance [31]. For these reasons, glutamine metabolism has been suggested as a promising target for ccRCC treatment [32,33]. Our results suggest, for the first time, a similar feature of glutamine addiction (higher consumption of glutamine as shown in Appendix A) in pazopanib-resistant cells. The importance of this amino acid in resistance mechanisms has prompted ongoing clinical trials to assess the efficacy of glutaminase inhibitors in RCC treatment [34,35].

Despite the common trends observed in the intracellular amino acid metabolism of resistant cells, a notable feature in the amino acid metabolism of pazopanib-resistant cells was the significant decrease in taurine levels. Taurine, classified as a non-essential amino acid, can be synthesised from methionine or obtained from the diet [36]. Previous work has shown that taurine acts as a protective substance with anti-tumour properties in various cancers, including breast [37], prostate [38] and colon-rectal [39]. This is due to its ability to suppress tumour cell proliferation, promote apoptosis, and induce autophagy [38]. The network analysis performed in this study revealed putative associations with the regulatory apoptotic genes *Casp3* and *BAX* (Table 1 and Appendix A). For these reasons, a down-regulation of taurine levels is expected to be associated with disease progression. Importantly, the decrease in taurine levels has been proposed as a potential biomarker for poor prognosis in colorectal cancer [39]. In addition, a previous metabolomic study showed lower taurine levels in leukaemia cell lines resistant to the TKI imatinib compared to parental cells [36]. As the metabolic mechanisms associated with the development of resistance to pazopanib remain unexplored, the observed reduction in taurine levels represents a potential direction for future studies, particularly as a strategy to overcome resistance to pazopanib in metastatic ccRCC. At the extracellular level, pazopanib-resistant cells displayed other unique characteristics, particularly in the increased consumption of various amino acids. In particular, arginine emerged as one of the most consumed amino acids by these cells, which is in line with the findings of the aforementioned paper by Hatakeyama et al. [18]. The authors highlight the important role of arginine in the regulation of chemoresistance, as this amino acid mediates immune cell function [40].

Our study also showed decreased intracellular levels of phosphocholine and glycerophosphocholine in both resistant cell lines, suggesting a dysregulation of glycerophospholipid metabolism. Glycerophospholipids are integral components of cellular membranes and energy reserves [41], and an increase in their levels would normally be expected to meet the high demands of cancer cell proliferation. Several studies have advocated the ChoP/GPC ratio as a useful biomarker of cancer cell growth [36,41,42]. Based on this, our results support the hypothesis of increased cell proliferation, as a significantly higher ChoP/GPC ratio was found in sunitinib-resistant cells (Appendix A). A similar trend was observed in pazopanib-resistant Caki-1 cells but did not reach statistical significance. Furthermore, the intracellular composition of sunitinib-resistant cells is characterised by an enrichment of phosphatidylethanolamines and their precursor ethanolamine. Since phosphatidylethanolamines are the major components of cell membranes [43], these results are consistent with the greater ability for cell proliferation that was observed in sunitinib-resistant cells.

The ^1^H NMR intracellular lipidome analysis performed in this study showed that both resistant cells shared a common increase in intracellular levels of monoglycerides. Sunitinib-resistant cells also showed higher levels of cholesteryl esters, whereas pazopanib-resistant cells showed a reduction in the intracellular levels of cholesterol, cholesteryl esters and fatty acids. Fatty acids are the major components of lipids and represent an extracellular requirement of solid tumours as a nutrient source [44]. Abnormal regulation of lipid metabolism is a recognised hallmark in the progression and development of drug resistance in ccRCC [31,32,44,45], highlighting the importance of investigating these metabolic pathways. Interestingly, a previous lipidomic study correlated altered fatty acid biosynthesis (e.g., accumulation of very long chain fatty acids and polyunsaturated fatty acids) with cisplatin resistance in tissue samples from ccRCC patients [46]. Unravelling the detailed lipid reprogramming mechanisms underlying sunitinib and pazopanib resistance has the potential to provide new insights into precise strategies to overcome these issues. In addition, previous transcriptomic data using sunitinib-resistant cells and ccRCC tissue have proposed a fatty acid metabolism-related risk model to assess the propensity for drug resistance in ccRCC [45].

Our results also pointed to disturbances in nicotinate and nicotinamide metabolism in both resistant RCC cell lines, as a common intracellular decrease in NAD^+^ was found. NAD^+^ is a cofactor in several critical processes for maintaining cellular homeostasis. Among its many roles, NAD^+^ plays an important role in regulating the Warburg effect, a metabolic phenomenon in which cancer cells switch to glycolysis rather than oxidative phosphorylation for energy requirements [47,48]. Dysregulation of nicotinate and nicotinamide metabolism has been reported by Hakimi et al. [49], who observed a decrease in related metabolites in ccRCC tissue compared to non-tumoral tissue. Thus, our observation for the first time demonstrates the importance of investigating NAD^+^ dysregulation in the context of TKI resistance in ccRCC. Importantly, NAD^+^ metabolism has been considered a promising therapeutic target for cancer treatment due to its role in promoting cancer cell proliferation and growth [50].

Specific metabolic changes in sunitinib-resistant cells include enhanced cellular antioxidant capacity, supported by a significant intracellular up-regulation of glutathione and myo-inositol, together with increased glutamine uptake, as described previously. Notably, glutathione and myo-inositol were found to be significantly elevated only in sunitinib-resistant cells. Glutathione plays a key role in maintaining intracellular oxidative balance, and myo-inositol is recognised as a vital growth-promoting factor for mammalian cells [51,52]. The enhanced antioxidant and proliferative capabilities confer an advantage on cancer cells by inhibiting oxidative stress-induced apoptosis [21]. Consequently, these mechanisms collectively contribute to the development of sunitinib resistance [18,23].

In pazopanib-resistant cells, unique and pronounced metabolic features suggest an impact on the regulation of energy acquisition processes. It is widely recognised that cancer cells use bioenergetic adaptations to fuel tumour progression and overcome drug-induced challenges [21]. The present analysis revealed decreased excretion of several organic acids involved in energy metabolism, including glycolysis/gluconeogenesis, TCA cycle and pyruvate metabolism. In addition to these metabolic changes, pazopanib-resistant cells exhibited significantly higher glucose consumption compared to parental cells. Glucose uptake plays a central role in the Warburg effect [53], providing the energy and substrates required for glycolysis and driving tumour progression [44]. The increased glycolytic rate leads to increased lactate production, contributing to an acidic cancer microenvironment that inhibits drug efficacy [44,54]. Indeed, an up-regulation of intracellular lactate levels was observed in resistant cells. The recognition of the role of Warburg effect mechanisms in chemoresistance has led to the initiation of clinical trials aimed at improving therapeutic regimens by targeting glycolysis transporters and key enzymes [48]. Despite the novelty of our findings regarding pazopanib resistance, several studies have reported an increased glucose uptake [55,56,57] and elevated lactate production [42,58,59] in various resistant cell lines, including those resistant to sunitinib [23]. In this study, only increased lactate excretion was observed in sunitinib-resistant cells, with no changes in the levels of other metabolites involved in energy metabolism. Network analysis of metabolite changes characteristic of pazopanib-resistant cells allowed the putative identification of associations between different genes related to pyruvate metabolism, TCA cycle and glycolysis. Indeed, specific related genes that may be associated with pazopanib resistance included *ABAT*, *AGXTs*, *IL4I1*, *APP* and *CASP3*.

This study is the first to explore the distinctive metabolic changes associated with sunitinib and pazopanib resistance and provides important insights to refine treatment strategies for metastatic ccRCC; however, as with any in vitro study, some limitations must be acknowledged. An inherent limitation of in vitro experiments is their inability to fully replicate the conditions in an organism, such as the influence of the tumour microenvironment and cellular interactions. These complexities may be better modelled in animal studies. However, in vitro models offer increased throughput and precise control of the chemical and physical environment, allowing the elimination of confounding factors present in more complex models. Therefore, investigation in animal models and validation in human samples from advanced-stage ccRCC patients treated with TKIs are essential for future studies. This study also lacks direct measurement of the targets of the TKIs under investigation (VEGFRs, PDGFRs, c-Kit) in parental and resistant cells. Nevertheless, a previous study has measured VEFG in Caki-1 cells [60] and showed an enrichment of its expression in sunitinib-resistant cells. Bioinformatics tools, such as pathway and network analysis, were used in the present study to infer putative changes in specific biochemical pathways and genes; however, these findings need to be validated and extended using genomic and proteomic approaches.

## 4. Materials and Methods

### 4.1. Materials for Cell Culture and Chemicals

RPMI-1640 medium, sunitinib malate (≥98%) and pazopanib (≥98%) were purchased from Sigma-Aldrich (Sigma-Aldrich, St. Louis, MO, USA). The antibiotic mixture penicillin/streptomycin (10,000 U/mL/10,000 mg/mL), heat-inactivated fetal bovine serum (FBS) and trypsin-EDTA (0.25%) were purchased from GIBCO Invitrogen (GIBCO, Barcelona, Spain). Deuterium oxide (D_2_O) and deuterated chloroform (CDCl_3_) with tetramethylsilane (TMS) were provided by Eurisotop (Eurisotop, Saint-Aubin, France). D_2_O containing 0.05 wt% 3-(trimethylsilyl)propionic-2,2,3,3-d4 acid (TSP) sodium salt was obtained from Sigma-Aldrich (Sigma-Aldrich, St. Louis, MO, USA). Thiazolyl blue tetrazolium bromide (MTT) and solvents such as dimethyl sulfoxide (DMSO), methanol and chloroform were purchased from Sigma-Aldrich (Sigma-Aldrich, St. Louis, MO, USA). Cell culture grade DMSO was purchased from PanReac (PanReac, Barcelona, Spain).

### 4.2. Cell Lines and Culture Conditions

The human metastatic ccRCC cell line Caki-1 was obtained from the American Type Culture Collection (ATCC; Manassas, VA, USA). Caki-1 cells were maintained in RPMI 1640 culture medium supplemented with 10% FBS and 1% penicillin/streptomycin in a humidified incubator at 37 °C under a 95% air/5% CO_2_ atmosphere. Caki-1 sublines resistant to the TKIs were generated based on previous literature [18,23,25,26,27]. Sunitinib- and pazopanib-resistant Caki-1 cell lines were established by continuously exposing the parental Caki-1 cells to gradually increasing concentrations of the respective drug for 6 months. The concentration ranges were 1, 2, 4, 6, 8, and 10 μM for sunitinib, and 1, 2, 4, 6, 12, 24, 36, 40, and 50 μM for pazopanib. Then, to ensure and maintain resistance for the subsequent experiments, 2 μM sunitinib or 24 μM pazopanib (roughly equivalent to EC_40_ values) were persistently retained in the culture medium of the respective resistant cell line. For each TKI drug, two resistant cell sublines were generated from different aliquots of the parental cell line. Parental cells were tested from passages 4 to 10, while the sunitinib-resistant Caki-1 cells were tested from passages 48 to 52, and pazopanib-resistant Caki-1 cells were tested from passages 58 to 64. All cell lines were routinely tested for *Mycoplasma* spp. contamination (TaKaRa PCR Mycoplasma Detection Set, Clontech Laboratories, Brussels, Belgium).

### 4.3. Cell Viability, Proliferation, and Morphological Assays

To confirm the resistance behaviour of the established resistant cell lines, the cell viability and growth rate of both parental and resistant cell lines were assessed under sunitinib or pazopanib exposure. Cell viability was measured using the MTT reduction assay as previously described [61]. Briefly, cells were initially seeded at a density of 15 × 10^3^ cells/well in 96-well culture plates. After 24 h of incubation, the culture medium was removed, and the cells were exposed to a fresh medium containing sunitinib (0.1, 0.2, 1, 2, 5, 10, 25, 50, and 100 µM) or pazopanib (0.1, 1, 5, 10, 25, 50, 100, and 200 µM), for 48 h. The medium was then aspirated, and 200 µL of MTT solution (0.5 mg/mL) was added to each well and incubated for 2 h at 37 °C in a 5% CO_2_ atmosphere. The MTT-containing medium was removed, and 100 μL of DMSO was added to each well to solubilise the purple MTT formazan crystals. Absorbance was measured at 550 nm using a microplate reader (BioTek Instruments, Winooski, VT, USA). Results were plotted as percentages of cell death relative to control from three independent experiments, with each concentration tested in triplicate within each experiment. The effective concentrations causing a 50% loss of cell viability (EC_50_ values) were calculated for each drug in the parental and drug-resistant cell lines. The resistance factor of the sunitinib- and pazopanib-resistant cell lines was calculated by dividing the EC_50_ values of the resistant cells by those of the parental cells. For the cell proliferation assay, parental and resistant Caki-1 cells were seeded at an initial density of 8 × 10^3^ cells/well in 96-well culture plates and allowed to adhere overnight. Cell growth was then measured using MTT assays performed immediately before (t = 0) and 24, 48, and 72 h after the addition of 2 μM sunitinib or 24 μM pazopanib. Experiments were performed in triplicate in at least three independent assays for each time point of analysis. Phase contrast microscopy was used to assess morphological changes induced by TKI resistance. Cell images were obtained using a Nikon Eclipse TS100 inverted microscope equipped with a DS-Fi1 camera (Nikon, Tokyo, Japan).

### 4.4. Metabolite Extraction and Sample Preparation for ^1^H NMR-Based Metabolomics

Parental and resistant Caki-1 cells were seeded at an initial density of 40 × 10^4^ cells/mL in 60 mm cell culture dishes and incubated at 37 °C with 5% CO_2_. Culture media without cells (blanks) were maintained under the same conditions. After a 24 h incubation, culture media from blanks, parental and resistant Caki-1 cells were collected and centrifuged (1200× *g*, 5 min, 4 °C), and the supernatant was stored at −80 °C for analysis of extracellular metabolites by ^1^H NMR spectroscopy. Cell monolayers were washed with PBS, and intracellular metabolites were extracted using a two-phase extraction method (methanol/chloroform/water 4:4:2.85) based on previous literature [62]. First, 800 µL of cold methanol was added to stop cellular metabolism. The cells were scrapped, transferred to a microcentrifuge tube, and sonicated on ice (3 × 30 s at 23 kHz and 10 μm amplitude using an exponential probe) for cell breakage. Then, 320 µL of chloroform (twice) and 320 µL of water were added. Samples were vortexed for 2 min between each solvent addition. After 10 min on ice, the samples were centrifuged (2000× *g*, 15 min plus 10,000× *g*, 2 min, 4 °C), and 700 µL of the upper phase (containing the polar metabolites) was collected, lyophilized, and stored at −80 °C until ^1^H NMR analysis. Similarly, 500 µL of the lower phase (containing lipids) was collected, dried under a nitrogen stream, and stored at −80 °C. Eight independent experiments from different passages were considered for each condition.

Sample preparation for ^1^H NMR analysis was based on the protocols recommended for in vitro metabolomics studies [63,64]. On the day of NMR analysis, 585 µL of culture medium was mixed with 65 µL of D_2_O containing 0.25% TSP, centrifuged (10,410× *g*, 5 min, 4 °C), and 600 µL of the supernatant was transferred to a 5 mm NMR tube. The dried intracellular polar extracts were resuspended in 650 µL of phosphate buffer (100 mM, pH 7.4, 100 µL % D_2_O) containing 0.25% TSP, centrifuged (10,410× *g*, 5 min, 4 °C) and 600 µL of the supernatant was transferred to a 5 mm NMR tube. Finally, the intracellular lipid extracts were resuspended in 650 µL of CDCl_3_ containing 0.015% TMS, vortexed, and 600 µL were transferred to the 5 mm NMR tubes.

### 4.5. ^1^H NMR-Based Metabolomic Analysis

The ^1^H NMR metabolic profiling was carried out at the Nuclear Magnetic Resonance Laboratory of the Materials Centre of the University of Porto (CEMUP). The experiments were performed on a Bruker Ascend 600 14.1 T spectrometer (Bruker BioSpin, Rheinstetten, Germany) operating at a frequency of 600 MHz for ^1^H and equipped with a BBO Cryogenic Prodigy probe (298 K). Standard one-dimensional (1D) ^1^H spectra (*noesypr1d* in the Bruker library) were recorded for dried intracellular polar extracts and culture medium supernatant with 4 s relaxation delay, 100 ms mixing time, 256 transients, 64 k data points, 10,080.646 Hz spectral width and 3.25 s acquisition time. The 1D spectra were processed with an exponential line broadening function (3.0 Hz for intracellular extracts and 1.0 Hz for culture medium), manually phased and baseline corrected, and chemical shifts referenced to TSP at δ = 0.00 ppm. The ^1^H NMR spectra of intracellular lipid extracts were acquired on a Bruker Avance III 400 9.4 T spectrometer (Bruker BioSpin, Rheinstetten, Germany) operating at 400 MHz (298 K) using the *zg* pulse sequence (Bruker library). The acquisition parameters were a 5 ms relaxation delay, 512 transients, 32 k data points, 7002.80 Hz spectral width, and 2.34 s acquisition time. Each free-induction decay was zero-filled to 64 k points and multiplied by a 1.0 Hz exponential line broadening function, followed by manual phase, baseline correction, and chemical shift calibration (TMS δ = 0.00 ppm).

### 4.6. Metabolite Annotation and ^1^H NMR Data Processing

For metabolite annotation, the ^1^H resonances present in the samples were compared with the ^1^H NMR spectra of standard compounds available in the Biological Magnetic Resonance Bank [65], the Human Metabolome Database [66], and the Chenomx NMR suite 8.4 software (Chenomx Inc., Edmonton, AB, Canada). Statistical total correlation spectroscopy (STOCSY) [67] was performed to aid in the annotation of intracellular lipids using the processed ^1^H NMR data and the ggplot2 package [68] in R 4.0.3 software [69]. In addition, existing literature [70] assisted in the annotation of monoglycerides and triglycerides in intracellular lipid extracts.

For data processing, ^1^H NMR spectra were uploaded to the NMRProcFlow 1.4 [71]. From the total spectra of polar intracellular extracts (9.5–0.5 ppm), the spectral regions of residual water (5.20–4.66 ppm), methanol (3.38–3.31 ppm), and DMSO (2.78–2.71 ppm) were excluded. For the processing of lipid extracts, a whole region of 8.25–0.5 ppm was considered, excluding the spectral regions of water (1.76–1.39 ppm), methanol (3.516–3.44 ppm), chloroform (7.69–6.88 ppm) and some impurities (1.155–1.06 and 4.914–4.79 ppm). For the culture medium, the entire spectral region considered was 9.5–0.5 ppm, and the spectral regions removed were those of residual ethanol (1.16–1.22 and 3.69–3.64 ppm), DMSO (2.80–2.65 ppm), and water (5.05–4.67 ppm). The spectra of intracellular extracts and culture medium were then aligned using the parametric time warping method [72], followed by uniform spectral width bucketing (0.001 ppm, signal-to-noise ratio = 3) and normalisation by the total area (TA).

### 4.7. Statistical Analysis and Biological Interpretation

Non-linear regression curves of cell viability assays and EC_50_ values were generated using GraphPad Prism (version 8.2.1, San Diego, CA, USA). For the proliferation assays, results were expressed as mean ± standard error of the mean and the statistical analysis was performed using GraphPad Prism (version 8.2.1, San Diego, CA, USA). Multiple comparisons were performed using a two-way analysis of variance (ANOVA) test, followed by Tukey’s post hoc test. Results were considered statistically significant when *p*-value *<* 0.05 (95% confidence level). Each ^1^H NMR matrix was uploaded to Metaboanalyst version 5.0 [73], scaled to unit variance (UV) and analysed using multivariate methods, including PCA and PLS-DA. A five-fold internal cross-validation was used to validate the performance of the PLS-DA models, determining the number of principal components (PC) and assessing the model’s accuracy (Acc), sum of squares (R^2^) and predictive ability (Q^2^) of the model. For data interpretation, PLS-DA loading plots were generated in R 4.0.3 software [68] using the ggplot2 package [69], and relevant resonances identified with VIP greater than 1.0 were considered for the Mann-Whitney test (GraphPad Prism, version 8.2.1, San Diego, CA, USA). *p*-values below 0.05 were considered statistically significant. The effect size (ES) and the corresponding standard error (SE) [74] were computed for metabolites showing significant statistical changes.

Pathway analysis was used to interpret the putatively affected metabolic pathways, and network analysis was used to propose putative changes in genes associated with these metabolic pathways. For this purpose, the set of statistically significant altered metabolites and lipids found in both intracellular and extracellular compartments of each resistant cell line was combined and uploaded to Metaboanalyst version 5.0 [73]. The Homo sapiens library from KEGG [75] was selected for pathway identification. A degree filter of 2.0 (indicating the number of direct connections a node has) and a betweenness cut-off of 1.0 (representing the betweenness centrality between other pairs of nodes) were chosen as parameters for network analysis. Significant network correlations were considered when the *p-*value was < 0.05.

## 5. Conclusions

Overall, our ^1^H NMR-based metabolomics analysis provides a comprehensive view of the metabolic rewiring associated with sunitinib and pazopanib resistance in metastatic ccRCC cells. Sunitinib- and pazopanib-resistant cancer cells undergo unique metabolic changes that facilitate growth, primarily by upregulating metabolites involved in antioxidant defence, energy, and lipid metabolism. Similarities observed in TKI-resistant cells included perturbations in the amino acid, glycerophospholipid, and nicotinate and nicotinamide metabolism. Importantly, drug-specific metabolic alterations were also found. Indeed, sunitinib-resistant cells showed increased antioxidant capacity, as evidenced by increased intracellular levels of glutathione and myo-inositol and increased glutamine uptake. In addition, these resistant cells showed increased cell proliferation, as evidenced by intracellular accumulation of lipids and glycerophospholipids. Pazopanib-resistant cells showed marked alterations in glycolysis/gluconeogenesis, tricarboxylic acid (TCA) cycle and pyruvate metabolism. This study highlights the importance of further investigation into the specific metabolic changes associated with resistance to sunitinib and pazopanib. These metabolic alterations are of remarkable importance in restoring cancer cell responsiveness to TKI therapy, thereby improving the prognosis of metastatic ccRCC patients.

## Figures and Tables

**Figure 1 ijms-25-06328-f001:**
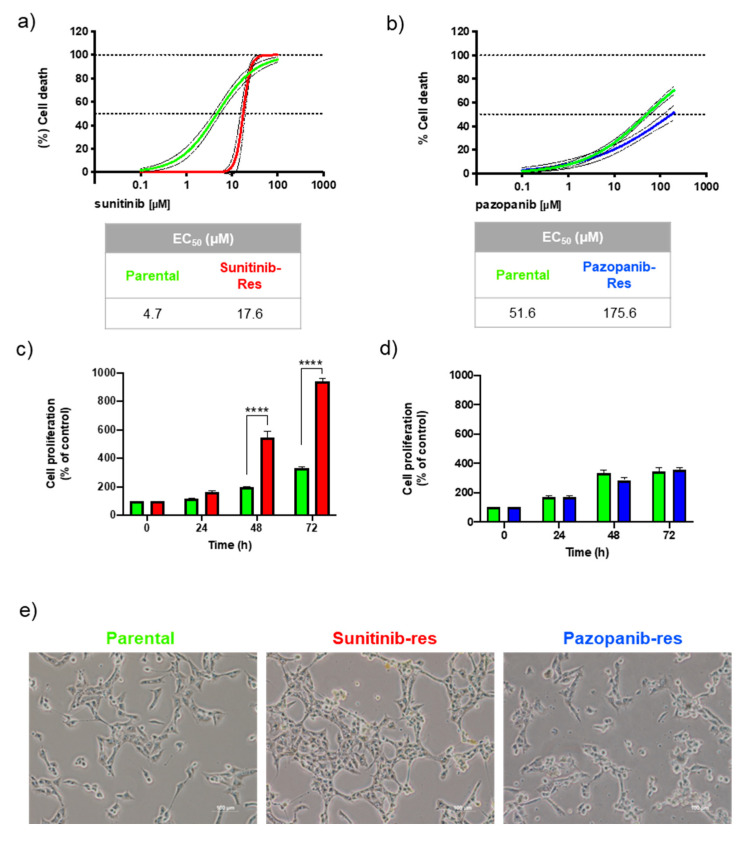
(**a**,**b**) Non-linear regression models represented with mean values and 95% confidence intervals obtained for sunitinib- and pazopanib-induced cell death in parental (green line), sunitinib-resistant (red line) and pazopanib-resistant (blue line) Caki-1 cell lines as assessed by the MTT after 48 h exposure. Statistical significance assessed between models using the Extra Sum of Squares F test revealing a *p* < 0.0001. (**c**,**d**) Effects on cell proliferation of parental (green bars), sunitinib-resistant (red bars) and pazopanib-resistant (blue bars) Caki-1 cells exposed to 2 µM sunitinib or 24 µM pazopanib. (**e**) Representative phase contrast microscopy images of parental and resistant Caki-1 cell lines. Scale bar: 100 µm. Original magnification 10×. Results were obtained from three independent experiments, performed in triplicate, and are presented as mean ± standard error of the mean. **** *p* < 0.0001.

**Figure 2 ijms-25-06328-f002:**
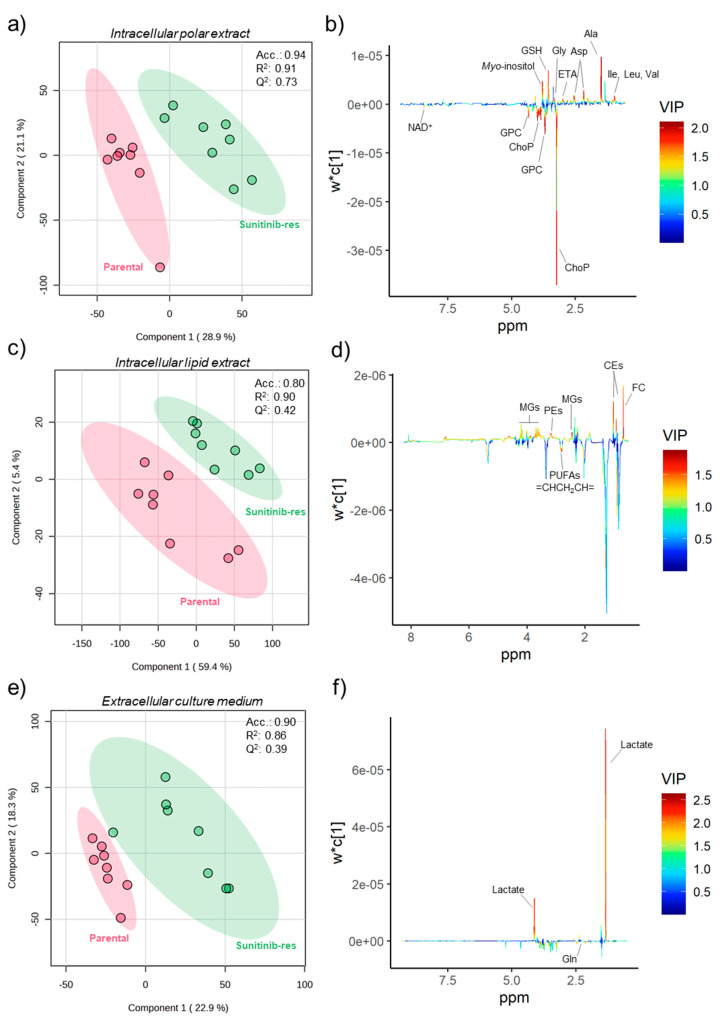
(**a**,**b**) PLS-DA score scatter and loading plots of intracellular polar metabolic profiles from sunitinib-resistant (green circles, *n* = 8) vs. parental (red circles, *n* = 8) Caki-1 cells. (**c**,**d**) PLS-DA score scatter and loading plots of intracellular lipid profiles from sunitinib-resistant (green circles, *n* = 8) vs. parental (red circles, *n* = 8) Caki-1 cells. (**e**,**f**) PLS-DA score scatter and loading plots of extracellular metabolic profiles from sunitinib-resistant (green circles, *n* = 8) vs. parental (red circles, *n* = 8) Caki-1 cell lines. Acc, R^2^ and Q^2^ values were obtained with two components. Abbreviations: Ala: alanine; Asp: aspartate; CEs: cholesteryl esters; ChoP: phosphocholine; ETA: ethanolamine; FC: free cholesterol; Gly: glycine; Gln: glutamine; GSH: glutathione; GPC: glycerophosphocholine; Iso: isoleucine; Leu: leucine; MGs: monoglycerides; NAD^+^: nicotinamide adenine dinucleotide; PEs: phosphatidylethanolamines; PUFAs: polyunsaturated fatty acids; Val: valine.

**Figure 3 ijms-25-06328-f003:**
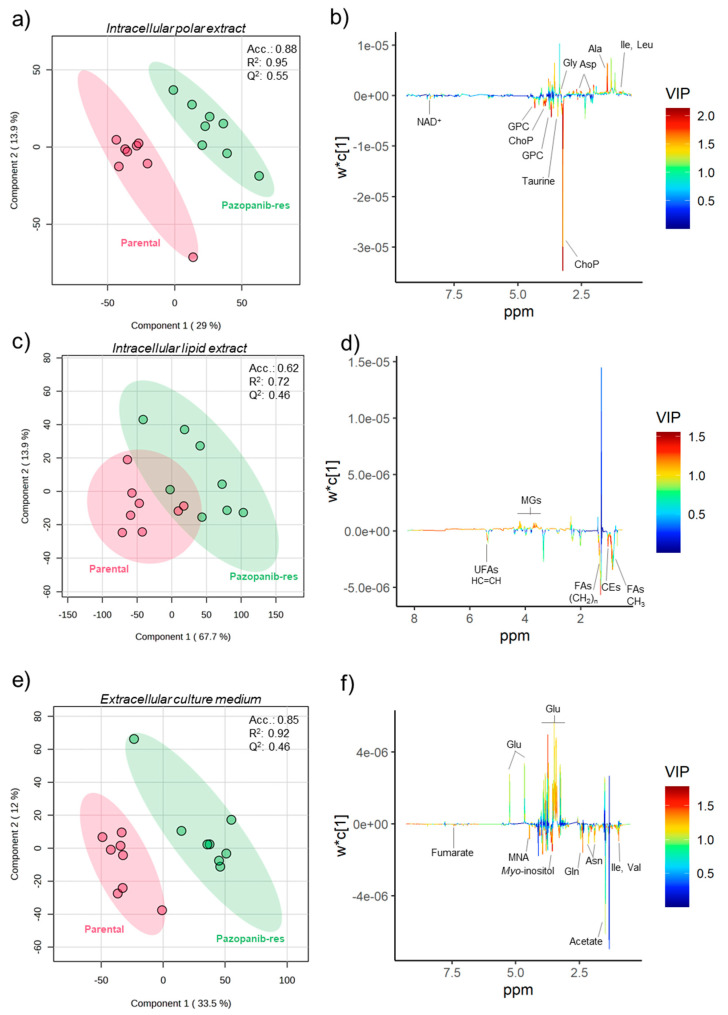
(**a**,**b**) PLS-DA score scatter and loading plots of intracellular polar metabolic profiles from pazopanib-resistant (green circles, *n* = 8) vs. parental (red circles, *n* = 8) Caki-1 cell lines. (**c**,**d**) PLS-DA scores scatter and loading plots of intracellular lipid profiles from pazopanib-resistant (green circles, *n* = 8) vs. parental (red circles, *n* = 8) Caki-1 cell lines. (**e**,**f**) PLS-DA scores scatter and loading plots of extracellular metabolic profiles from pazopanib-resistant (green circles, *n* = 8) vs. parental (red circles, *n* = 8) Caki-1 cell lines. Acc, R^2^ and Q^2^ values were obtained with two components. Abbreviations: Ala: alanine; Asn: asparagine; Asp: aspartate; CEs: cholesteryl esters; ChoP: phosphocholine; FAs: fatty acids: Gly: glycine; Gln: glutamine; Glu: glucose; GPC: glycerophosphocholine; Iso: isoleucine; Leu: leucine; MGs: monoglycerides; MNA: 1-methylnicotinamide; NAD^+^: nicotinamide adenine dinucleotide; UFAs: unsaturated fatty acids; Val: valine.

**Figure 4 ijms-25-06328-f004:**
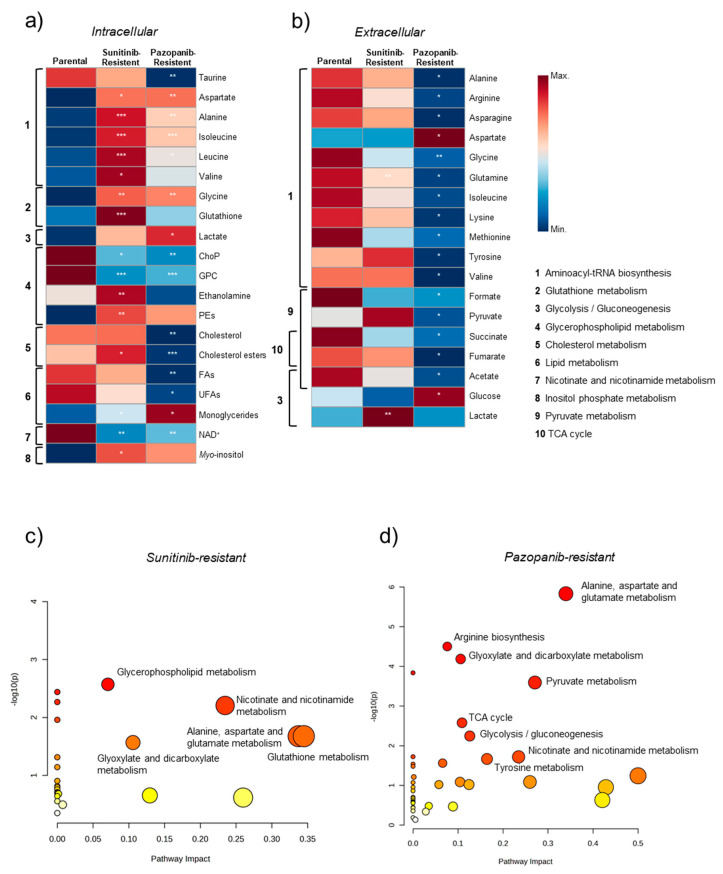
(**a**,**b**) Heatmaps illustrating the mean levels of intracellular and extracellular metabolites altered in sunitinib- and pazopanib-resistant Caki-1 cell lines and the putatively altered metabolic pathways. Columns represent each sample group, and rows correspond to the mean normalised peak area of each metabolite coloured from minimum value (dark blue) to maximum value (dark red). Statistical significance was assessed by comparison with the parental Caki-1 cell line (first column in each heatmap) * *p* < 0.05; ** *p* < 0.01; *** *p* < 0.001. (**c**,**d**) Pathway analysis was performed on the list of metabolites found to be altered in sunitinib- and pazopanib-resistant Caki-1 cell lines, respectively. The annotated pathways were considered statistically significant (*p* < 0.05). Abbreviations: ChoP: phosphocholine; FAs: fatty acids; PEs: phosphatidylethanolamine; NAD^+^: nicotinamide adenine dinucleotide; UFAs: unsaturated fatty acids.

**Figure 5 ijms-25-06328-f005:**
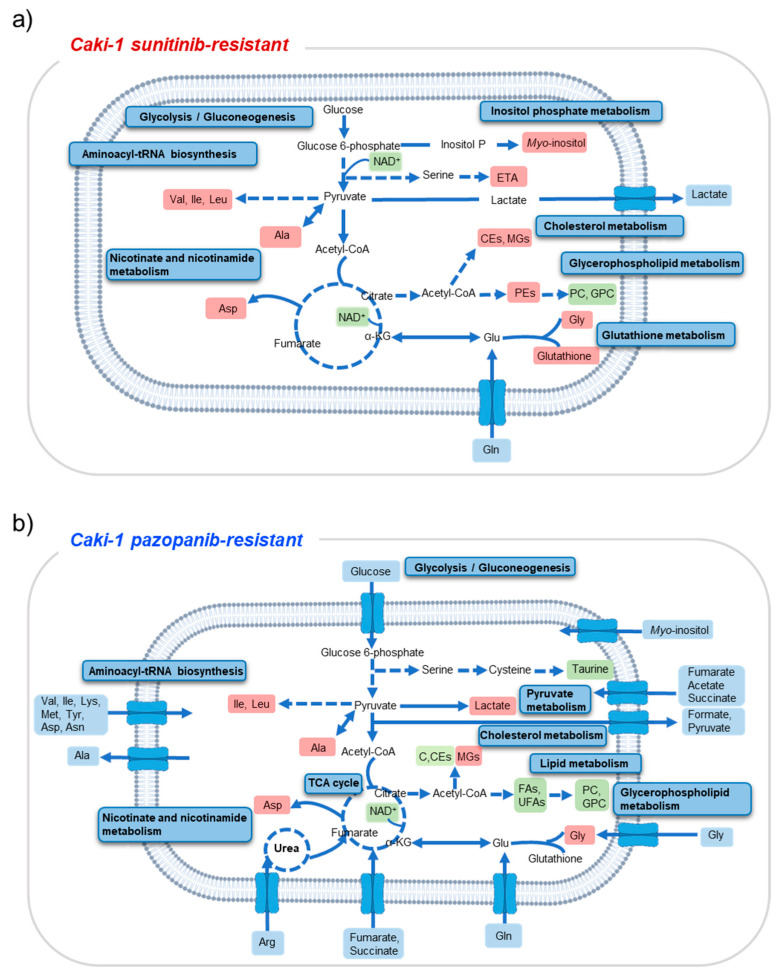
(**a**,**b**) Putative metabolic changes in sunitinib- and pazopanib-resistant Caki-1 cells. Red and green squares indicate increased and decreased metabolites, respectively. Blue squares represent extracellular changes. Arrows indicate metabolites consumed (orientation from outside to inside) and excreted (orientation from inside to outside). The consumption and excretion of metabolites can be interpreted from the boxplots shown in Appendix A. Dashed lines represent multiple-step reactions. Abbreviations: Ala: alanine; Arg: arginine; Asn: asparagine; Asp: aspartate; C: cholesterol; CEs: cholesteryl esters; ChoP: phosphocholine; ETA: ethanolamine; FAs: fatty acids; Gly: glycine; Gln: glutamine; Glu: glutamate; GSH: glutathione; GPC: glycerophosphocholine; Iso: isoleucine; Leu: leucine; Lys: lysine; Met: methionine; MGs: monoglycerides; NAD^+^: nicotinamide adenine dinucleotide; PEs: phosphatidylethanolamines; TCA: tricarboxylic acid cycle; Tyr: tyrosine; UFAs: unsaturated fatty acids; Val: valine.

**Table 1 ijms-25-06328-t001:** List of putative metabolic pathways, metabolites and related genes associated with sunitinib- and pazopanib-resistant Caki-1 cells generated by network analysis.

Dysregulated Metabolic Pathway	Metabolites	Relative-Related Genes	*p*-Value
**Sunitinib-resistant Caki-1**			
Aminoacyl-tRNA biosynthesis	Glycine, aspartate, glutamine, alanine, isoleucine, leucine, valine	*DLD*, *SLC7A8*, *GPRC6A*, *SLC38A5*, *LDHA*, *PKM2*, *EPRS*	<0.0001
Valine, leucine, and isoleucine biosynthesis	Isoleucine, leucine, valine	*IARS*, *KARS*	<0.0001
Alanine, aspartate, and glutamate metabolism	Aspartate, glutamine, alanine	*ALAD*, *SLC38A2*, *PKLR*, *LYZ*, *GPT2*	<0.0001
Valine, leucine, and isoleucine degradation	Isoleucine, leucine, valine	*LAP3*, *ALAD*, *AIMP1*, *ARID4B*, *IARS*, *KARS*	<0.0001
Arginine biosynthesis	Aspartate, glutamine	*SLC38A2*, *PKLR*, *LYZ*, *GPT2*	<0.0001
Pantothenate and CoA biosynthesis	Aspartate, valine	*IARS*, *KARS*	0.00135
Glutathione metabolism	Glycine, glutathione	*GRIN2B*	0.0479
Glyoxylate and dicarboxylate metabolism	Glycine, glutamine	*LAP3*	0.0479
**Pazopanib-resistant Caki-1**			
Aminoacyl-tRNA biosynthesis	Glycine, aspartate, tyrosine, glutamine, arginine, alanine, lysine, methionine, isoleucine, leucine, valine, asparagine	*LAP3*, *SLC7A8*, *GPRC6A*, *F2*, *MARS*, *PKLR*, *DECR1*, *TAC1*, *LDHA*, *AIMP2*	<0.0001
Alanine, aspartate, and glutamate metabolism	Aspartate, glutamine, succinate, alanine, pyruvate, asparagine	*BAX*, *MAPT*, *GAD1*, *CTH*, *GLYAT*, *GRIN2B*, *DLD*, *SLC6A19*, *GATM*, *ASNS*, *ASLl*, *PKM2*, *EPRS*, *PC*	<0.0001
Arginine biosynthesis	Aspartate, glutamine, arginine	*BAX*, *SLC6A19*, *GATM*, *ASNS*, *PKM2*, *EPRS*, *PC*, *ARID4B*, *IARS*	<0.0001
Glyoxylate and dicarboxylate metabolism	Glycine, glutamine, pyruvate	*CAT*, *BAX*, *ABAT*, *CTH*, *CASP3*	0.00012
Valine, leucine, and isoleucine biosynthesis	Isoleucine, leucine, valine	*LDHAL6A*, *LDHAL6B*	<0.0001
Pyruvate metabolism	Pyruvate, lactate	*ABAT*, *AGXT*, *IL4I1*, *SLC6A18*, *AGXT2*, *SLC38A5*, *SLC16A10*, *SLC6A14*, *APP*, *CASP3*	<0.0001
Citrate cycle (TCA cycle)	Succinate, pyruvate	*ABAT*, *AGXT2*, *CASP3*	0.0052
Glycolysis/Gluconeogenesis	Succinate, pyruvate	*ABAT*, *AGXT*, *IL4I1*, *SLC6A18*, *SLC38A5*, *SLC16A10*, *SLC6A14*, *APP*	<0.0001
Valine, leucine, and isoleucine degradation	Isoleucine, leucine, valine	*GAD1*, *ABAT*, *GLYAT*, *GRIN2B*, *KARS*, *LDHB*, *LDHC*, *LDHAL6A*, *LDHAL6B*	<0.0001
Pantothenate and CoA biosynthesis	Aspartate, valine	*LDHAL6A*, *LDHAL6B*	0.013

The *p*-value represents the statistical significance of the correlation between a metabolite and its relative related genes within metabolic pathways.

## Data Availability

The original contributions presented in the study are included in the article/Appendix A. Further inquiries can be directed to the corresponding author/s.

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
