# Peer review of "Metabolomics Reveals Tyrosine Kinase Inhibitor Resistance-Associated Metabolic Events in Human Metastatic Renal Cancer Cells"

_ijms, 2024, doi:10.3390/ijms25126328_

Round 1

Reviewer 1 Report

Comments and Suggestions for Authors

In the manuscript "Metabolomics reveals mechanisms of tyrosine kinase inhibitor resistance in renal cell carcinoma", a team of investigators led by Amaro Pinto describes the mechanisms of tyrosine kinase inhibitor resistance from a metabolomics perspective. The manuscript also highlights the significantly involved target genes and signaling pathways.

Although there were several interesting points made in this manuscript, it suffers from some weaknesses that need to be addressed before it is accepted.

1. Since the author introduces the VHL gene inactivation contributes to the ccRCC. The author needs to measure the expression of VEGF, PDGF, and c-Kit by western blot or flow. Suppose pazopanib can not target that marker in a well-established cell line.

2. The author needs to perform in multiple cell lines. More function assays are needed to evaluate the established resistance cell line. In addition to the proliferation assay, what about the apoptosis assay in the established resistance cell lines? 

3. Low resolution in Figures 4C and 4D, and hard to read. Please improve it.

4. At least some target gene expression involved in this resistance cell line needed to be evaluated, such as RT-PCR, or wb.

Author Response

Reviewer 1

Comments and Suggestions for Authors

In the manuscript "Metabolomics reveals mechanisms of tyrosine kinase inhibitor resistance in renal cell carcinoma", a team of investigators led by Amaro Pinto describes the mechanisms of tyrosine kinase inhibitor resistance from a metabolomics perspective. The manuscript also highlights the significantly involved target genes and signaling pathways.

Although there were several interesting points made in this manuscript, it suffers from some weaknesses that need to be addressed before it is accepted.

Author's reply: Thank you for your constructive and insightful comments on our article. We appreciate the positive feedback and are committed to addressing the issues raised to improve the quality of our manuscript. Below is our response to each point. We would like to emphasise that the lines mentioned in the responses correspond to those in the manuscript marked with track changes.

  1. Since the author introduces the VHL gene inactivation contributes to the ccRCC. The author needs to measure the expression of VEGF, PDGF, and c-Kit by western blot or flow. Suppose pazopanib can not target that marker in a well-established cell line.

Authors’ reply: We would like to thank Reviewer 1 for this comment. We understand the importance of measuring the expression of VEGF, PDGF, and c-Kit, but performing additional experiments is beyond the current resources available for this study. A previous study measured VEGF expression in Caki-1 cells and showed higher levels in sunitinib-resistant cells (doi: 10.1016/j.neo.2015.11.001). Regarding pazopanib, a previous study supports its efficacy in targeting these markers in ccRCC cells (Caki-2 and 786-O) (doi: 10.1177/2041731420920597). Indeed, we have highlighted this limitation in the discussion section of our manuscript.

Page 25, lines 463 - 469: “This study also lacks direct measurement of the targets of the TKIs under investigation (VEGFRs, PDGFRs, c-Kit) in parental and resistant cells. Nevertheless, a previous study has measured VEFG expression in Caki-1 cells [60] and showed an enrichment of its expression in sunitinib-resistant cells. Bioinformatics tools, such as pathway and network analysis, were used in the present study to infer putative changes in specific biochemical pathways and genes; however, these findings need to be validated and extended using genomic and proteomic approaches.”

New reference:

Han, K. S., Raven, P. A., Frees, S., Gust, K., Fazli, L., Ettinger, S., Hong, S. J., Kollmannsberger, C., Gleave, M. E., & So, A. I. (2015). Cellular Adaptation to VEGF-Targeted Antiangiogenic Therapy Induces Evasive Resistance by Overproduction of Alternative Endothelial Cell Growth Factors in Renal Cell Carcinoma. Neoplasia (New York, N.Y.), 17(11), 805–816.

  1. The author needs to perform in multiple cell lines. More function assays are needed to evaluate the established resistance cell line. In addition to the proliferation assay, what about the apoptosis assay in the established resistance cell lines? 

Authors’ reply: We thank Reviewer 1 for this comment. We have attempted to induce resistance in other ccRCC cell lines (769-P and 786-O) without success. Inducing resistance in cancer cells is challenging and requires prolonged exposure to the drug. As a key advantage of our work, the ccRCC cell line that successfully developed TKI-resistance stands out for its metastatic potential, which more closely mimics clinical conditions when sunitinib and pazopanib are used in advanced stages. Our proliferation assays suggested a cytostatic effect rather than a cytotoxic effect, and together with the measurement of EC50 and the visualisation of morphological changes in the cells, we believe this is sufficient to indicate resistance. This has been previously reported by other authors (doi: 10.3390/ijms22126467, 10.15586/jkcvhl.2018.106, 10.3390/cancers12113150, 10.1007/s13402-015-0218-8).

  1. Low resolution in Figures 4C and 4D, and hard to read. Please improve it.

Authors’ reply: We would like to thank Reviewer 1 for this observation. We have improved the resolution of figures 4C and 4D. The updated figures have been included in the revised manuscript.

  1. At least some target gene expression involved in this resistance cell line needed to be evaluated, such as RT-PCR,or wb.

Authors’ reply: We would like to thank Reviewer 1 for the suggestions. As mentioned in our response to comment 1, we understand the importance of this assessment. We have highlighted this limitation in the discussion section of our manuscript and will consider it in future studies.

Reviewer 2 Report

Comments and Suggestions for Authors

The study examined the drug resistance-related metabolic changes in Caki-1 cell, which is a cell line derived from a human clear cell renal cell carcinoma (ccRCC), after the treatments of sunitinib and pazopanib, two tyrosine kinase inhibitors. The main technical platform is NMR-based metabolomics. There are some concerns on the quality of data, data interpretation, and discussion that need to be addressed by additional analysis and revisions.

---Some data on the TKI-elicited cell death and proliferation (Fig. 1) are puzzling or incomplete. In Fig. 1a, there is a very narrow window in the dose response curve that changed the % cell death from 0% to 100% (most likely occurred in the transition from 25 um to 50 uM), making the curve looking abnormal. In addition, no standard deviation and statistics are shown in the cell death plots (Fig. 1a-b) while they are present in the cell proliferation plot (Fig. 1c-d). Please explain and re-perform the experiments as needed (such as adding the concentrations between 0-100% death in the sunitinib experiment).

---On metabolomics data analysis, please consider to perform the correlation analysis between intracellular metabolites and extracellular metabolites, to demonstrate whether there are reciprocal relations on individual metabolites. This will help the discussion and data interpretation.

---Some issues on the scientific accuracy of summary figures (Fig. 5a-b). 1)pyruvate to acetyl-CoA occurs inside the mitochondrion.  2) the arrow from pyruvate to branched-chain amino acids (Val, Leu, Ile) could not be correct as these AAs are essential AAs, not from pyruvate. 3) Ethanolamine (ETA) is from serine in its biosynthesis. What is the arrow from acetyl-CoA to ETA? Please check them and others carefully.

---The current title  "Metabolomics reveals mechanisms of tyrosine kinase inhibitor 2 resistance in renal cell carcinoma" is not an appropriate statement. There is no validated mechanistic info in this manuscript since no additional gene and protein analysis was conducted in this study to explore the biochemical mechanisms of observed metabolic changes. A title such as "Identification of tyrosine kinase inhibitor resistance-associated metabolic events in renal cell carcinoma through metabolomic analysis" might be more appropriate.

---What are the p-values in Table 1? The current statement on line 603-604 is not self-explanatory.

---Please provide a paragraph on the limitation of this type of cell-based study (versus animal model) since the ccRCC cells are not the sole targets of TKIs (such as angiogenesis and other known mechanisms, which can not be explained by the ccRCC cell treatment).

Comments on the Quality of English Language

More careful proofreading is needed.

Author Response

Reviewer 2

Comments and Suggestions for Authors

The study examined the drug resistance-related metabolic changes in Caki-1 cell, which is a cell line derived from a human clear cell renal cell carcinoma (ccRCC), after the treatments of sunitinib and pazopanib, two tyrosine kinase inhibitors. The main technical platform is NMR-based metabolomics. There are some concerns on the quality of data, data interpretation, and discussion that need to be addressed by additional analysis and revisions.

Author's reply: Thank you for your constructive and insightful comments on our article. We appreciate the positive feedback and are committed to addressing the issues raised to improve the quality of our manuscript. Below is our response to each point. We would like to emphasise that the lines mentioned in the responses correspond to those in the manuscript marked with track changes.

  1. Some data on the TKI-elicited cell death and proliferation (Fig. 1) are puzzling or incomplete. In Fig. 1a, there is a very narrow window in the dose response curve that changed the % cell death from 0% to 100% (most likely occurred in the transition from 25 um to 50 uM), making the curve looking abnormal. In addition, no standard deviation and statistics are shown in the cell death plots (Fig. 1a-b) while they are present in the cell proliferation plot (Fig. 1c-d). Please explain and re-perform the experiments as needed (such as adding the concentrations between 0-100% death in the sunitinib experiment).

Authors’ reply: We thank the Reviewer 2 for this comment. We have already improved the curves to include the standard deviation along with the 95% confidence intervals, as shown on page 5.

Page 7, lines 133-137: “Figure 1. (a, b) Non-linear regression models represented with mean values and 95% confidence intervals obtained for sunitinib- and pazopanib-induced cell death in parental (green line), sunitinib-resistant (red line) and pazopanib-resistant (blue line) Caki-1 cell lines as assessed by the MTT assay after 48 h exposure. Statistical significance assessed between models using the Extra Sum of Squares F test revealing a p < 0.0001.”

  1. On metabolomics data analysis, please consider to perform the correlation analysis between intracellular metabolites and extracellular metabolites, to demonstrate whether there are reciprocal relations on individual metabolites. This will help the discussion and data interpretation.

Authors’ reply: We thank Reviewer 2 for this comment. We performed a correlation analysis between the intracellular and extracellular metabolites using the Spearman's correlation. However, due to the limited sample size (n = 8), we did not obtain robust correlations (correlation coefficients were below |0.5|). To provide additional context regarding metabolite consumption and excretion, we compared the extracellular culture medium of resistant cells with blanks (extracellular medium without cells).

  1. Some issues on the scientific accuracy of summary figures (Fig. 5a-b). 1)pyruvate to acetyl-CoA occurs inside the mitochondrion. 2) the arrow from pyruvate to branched-chain amino acids (Val, Leu, Ile) could not be correct as these AAs are essential AAs, not from pyruvate. 3) Ethanolamine (ETA) is from serine in its biosynthesis. What is the arrow from acetyl-CoA to ETA? Please check them and others carefully.

Authors’ reply: We would Reviewer 2 for this comment. We have carefully reviewed the figures and provided an improved version on page 16.

  1. The current title "Metabolomics reveals mechanisms of tyrosine kinase inhibitor 2 resistance in renal cell carcinoma" is not an appropriate statement. There is no validated mechanistic info in this manuscript since no additional gene and protein analysis was conducted in this study to explore the biochemical mechanisms of observed metabolic changes. A title such as "Identification of tyrosine kinase inhibitor resistance-associated metabolic events in renal cell carcinoma through metabolomic analysis" might be more appropriate.

Authors’ reply: We would like to thank Reviewer 2 for this suggestion with which we agree. We have changed it to: “Metabolomics reveals tyrosine kinase inhibitor resistance-associated metabolic events in renal cell carcinoma”, as we believe this clearly reflects the topic of the article.

  1. What are the p-values in Table 1? The current statement on line 603-604 is not self-explanatory.

Authors’ reply: p-Values in Table 1 represent the statistical significance of the correlation between a metabolite and its relative related-genes within metabolic pathways, as determined by network analysis.

We have added this information as footnote in Table 1:

Page 17: “The p-value represents the statistical significance of the correlation between a metabolite and its relative related-genes within metabolic pathways.”

We have also improved the information in “Materials and Methods section:

Page 29, lines 620-621: “Significant network correlations were considered when the p-value was below 0.05.”

---Please provide a paragraph on the limitation of this type of cell-based study (versus animal model) since the ccRCC cells are not the sole targets of TKIs (such as angiogenesis and other known mechanisms, which can not be explained by the ccRCC cell treatment).

Authors’ reply: We would like to thank Reviewer 2 for this suggestion. We have added a paragraph to explain the limitations of in vitro studies compared to in vivo animal models.

Page 25, lines 457-463: “An inherent limitation of in vitro experiments is their inability to fully replicate the conditions in an organism, such as the influence of the tumour microenvironment and cellular interactions. These complexities may be better modelled in animal studies. However, in vitro models offer increased throughput and precise control of the chemical and physical environment, allowing the elimination of confounding factors present in more complex models. Therefore, investigation in animal models and validation in human samples from advanced stage ccRCC patients treated with TKIs are essential for future studies.”

Reviewer 3 Report

Comments and Suggestions for Authors

Authors did lots of work studying the putative mechanisms of sunitinib-resistant and pazopanib-resistant in renal cell carcinoma cells (RCCs). Untargeted metabolomics is performed to comparing intra and extra metabolome of RCCs between resistant and sensitive RCCs, which is beneficial to elucidate the mechanisms of medicine resistance and provide potential biomarkers for the therapy of medicine resistant RCCs. Here are some suggestions for the manuscript:

1.      Fort the overlapping and unique metabolites between intra- and extra-cells, authors can consider adding a Venn diagram to demonstrate them.

2.      Is there any validation for the results? For example, the validation for metabolites’ identification/quantification/pathways, like purchasing standard metabolites for validation.

3.      Authors mentioned in Hatakeyama’s publication, sunitinib-resistant RCC cell line (786-O cell line), showed a significant increase in the intracellular levels of several amino acids (AAs). Please specify the AAs found in their study.  Are authors findings consistent with the results in Hatakeyama’s paper? Please add more details in the discussion section. If authors found more AAs, they would be the highlights of this paper.

4.      Please deposit the raw files to public resources and include the accession number into the manuscript.

Author Response

Reviewer 3

Comments and Suggestions for Authors

Authors did lots of work studying the putative mechanisms of sunitinib-resistant and pazopanib-resistant in renal cell carcinoma cells (RCCs). Untargeted metabolomics is performed to comparing intra and extra metabolome of RCCs between resistant and sensitive RCCs, which is beneficial to elucidate the mechanisms of medicine resistance and provide potential biomarkers for the therapy of medicine resistant RCCs. Here are some suggestions for the manuscript:

Author's reply: Thank you for your constructive and insightful comments on our article. We appreciate the positive feedback and are committed to addressing the issues raised to improve the quality of our manuscript. Below is our response to each point. We would like to emphasise that the lines mentioned in the responses correspond to those in the manuscript marked with track changes.

  1. Fort the overlapping and unique metabolites between intra- and extra-cells, authors can consider adding a Venn diagram to demonstrate them.

Authors’ reply: We would like to thank Reviewer 3 for the suggestion. We have added a Venn diagram to Figure S1c in Supplementary Materials.

Figure S1. (c) Venn diagram illustrating the number of unique and shared metabolites between the intracellular polar extract and the extracellular culture medium of Caki-1 cells.

  1. Is there any validation for the results? For example, the validation for metabolites’ identification/quantification/pathways, like purchasing standard metabolites for validation.

Authors’ reply: We thank reviewer 3 for this question. As mentioned in the “Materials and Methods” section (page 28, 574-576), metabolite annotation was performed by comparing the 1H resonances present in the samples with the 1H NMR spectra of standard compounds available in databases.

For pathway analysis, we used bioinformatics tools to infer putative changes in pathways. We have acknowledged the importance of validating these results in future studies and have added a sentence stating this limitation.

Page 25, line 466-469: “Bioinformatics tools, such as pathway and network analysis, were used in the present study to infer putative changes in specific biochemical pathways and genes; however, these findings need to be validated and extended using genomic and proteomic approaches.”

  1. Authors mentioned in Hatakeyama’s publication, sunitinib-resistant RCC cell line (786-O cell line), showed a significant increase in the intracellular levels of several amino acids (AAs). Please specify the AAs found in their study. Are authors findings consistent with the results in Hatakeyama’s paper? Please add more details in the discussion section. If authors found more AAs, they would be the highlights of this paper.

Authors’ reply: We would like to thank Reviewer 3 for this suggestion. Our results are consistent with previous observations from the Hatakeyama study, which reported increased levels of glycine, tyrosine, valine, isoleucine, aspartate, and glutamate in sunitinib-resistant cell lines. We have clarified this information in the revised manuscript. No other amino acids were found to be related to sunitinib resistance in our study.

Page 22, lines 346-347: “Consistent with these results, a previous study conducted by Hatakeyama et al. [18] investigating metabolic changes in a sunitinib-resistant RCC cell line (786-O cell line) showed a significant increase in the intracellular levels of several amino acids, including glycine, tyrosine, valine, isoleucine, aspartate, and glutamate.”

  1. Please deposit the raw files to public resources and include the accession number into the manuscript.

Authors’ reply: We would like to thank Reviewer 3 for the suggestion. As stated in the "Data availability statement", the raw datasets are included in the supplementary materials. We have added a sentence to the “Supplementary Materials” section for clarity.

Page 231, line 667-668: Dataset file: 1H NMR matrices of intracellular polar and lipid extracts and culture media.

Round 2

Reviewer 2 Report

Comments and Suggestions for Authors

The revisions have addressed my comments. As for the title, I recommend to use "human renal cancer cell", instead of "renal cell carcinoma", since it is an in vitro cell line study, not an in vivo carcinoma study.

Comments on the Quality of English Language

It is fine.

Author Response

We thank the reviewer for this comment. We have changed the title to "Metabolomics reveals tyrosine kinase inhibitor resistance-associated metabolic events in human metastatic renal cancer cells".